# Rapid Identification of Easily-Confused Mineral Traditional Chinese Medicine (TCM) Based on Low-Wavenumber Raman and Terahertz Spectroscopy

Meilan Ge [1,2], Yuye Wang [1,2,*], Haibin Li [1,2], Yu Wang [1,2], Changhao Hu [1,2], Junfeng Zhu [3], Degang Xu [1,2], Bin Wu [3] and Jianquan Yao [1,2]

1   Institute of Laser and Optoelectronics, School of Precision Instruments and Optoelectronics Engineering, Tianjin University, Tianjin 300072, China; meilange@tju.edu.cn (M.G.); haibin_li@tju.edu.cn (H.L.); 2020202055@tju.edu.cn (Y.W.); huchh97@tju.edu.cn (C.H.); xudegang@tju.edu.cn (D.X.); jqyao@tju.edu.cn (J.Y.)
2   Key Laboratory of Optoelectronic Information Technology (Ministry of Education), Tianjin University, Tianjin 300072, China
3   Science and Technology on Electronic Test & Measurement Laboratory, China Electronics Technology Group Corporation, Qingdao 266555, China; zhujfeng@mail.ustc.edu.cn (J.Z.); wubin@ei41.com (B.W.)
*   Correspondence: yuyewang@tju.edu.cn

**Abstract:** With the unique advantages of mineral TCMs gradually emerging in clinical treatment, health care, and precaution, it has played an important role in the international medical market. Commonly, mineral TCMs with similar appearance and different processing methods have different effects, but they are easy to be confused in preparation, storage, transportation, and other links, which affects the use and causes related problems. In this paper, six kinds of easily confused mineral TCMs, including coral skeleton, ophicalcitum, calamine, matrii sulfas exsiccatus, gypsum, and alumen, are rapidly characterized using Raman spectroscopy, which can be distinguished with different Raman peaks at 0–300 cm$^{-1}$ due to the different lattice structure. The THz spectra of these mineral TCMs show that different mineral TCMs have different THz absorption coefficients at 0.3–2.0 THz. Furthermore, compared with the ineffectiveness of the Raman spectrum for differentiating mineral TCMs prepared with disparate processing methods, the terahertz absorption spectrum plays an active role in making up the limitation of low-wavenumber Raman spectroscopy. The combination of low-wavenumber Raman and THz spectroscopy provides a simple and feasible scheme for the identification of mineral TCMs, which could play an important role in the quality control of mineral TCMs.

**Keywords:** mineral traditional Chinese medicine; low-wavenumber Raman spectroscopy; terahertz spectroscopy; identification

## 1. Introduction

Traditional Chinese medicine (TCM) has a long history in disease prevention and control. Since its successful application in the treatment of COVID-19, TCM is playing an increasingly significant role in the international medical field [1,2]. Thus, quality control is critically important for the safety and efficacy of TCM. Mineral TCM, as a part of TCM, has been widely used in clinical treatment. The types of mineral TCMs are very diverse and divided by the main components. Some kinds of mineral TCM with similar appearances or different processing methods have totally different clinical therapeutic effects, but they are easy to be confused in practice, such as storage and transportation [3]. Until now, the identification of mineral TCMs has mainly relied on morphological identification with or without microscopy. Even experienced technicians cannot identify samples that are similar in appearance. Until now, many other techniques have been employed to distinguish mineral TCMs. XRD can obtain information about the structure of atoms

or molecules inside samples by diffraction patterns, but it is expensive, complex, and requires expertise [4]. The ethylenediaminetetraacetic acid (EDTA) titration method can be adapted to determine the mineral ingredients according to standard solutions. However, the operation is very complicated and time-consuming [5]. Considering that near-infrared (NIR) spectroscopy can derive characteristic information of chemical functional groups in materials, it has also been used to distinguish mineral TCMs. However, it still displays some shortcomings, such as being highly sensitive to the effect of water and presenting difficulty with data modeling [6,7]. Therefore, it is highly desirable to establish a rapid and accurate method for the identification and quality control of mineral TCMs.

Raman spectroscopy is based on the inelastic scattering of photons by molecules, which has been widely applied in analyzing and identifying intramolecular vibration and/or rotation information of various compounds [8]. As an effective supplement to the traditional method, Raman spectroscopy is an ideal technique for identifying easily confused mineral TCMs due to its rapid, label-free, and non-destructive characteristics. It has been demonstrated that some kinds of mineral TCM containing sulfate [9] and $CaCO_3$ [10] can be characterized using Raman spectroscopy based on cluster analysis and synergy interval partial least squares. Because the same anionic group has similar Raman characteristic peaks, different samples attributed to mineral salt TCMs with the same anionic group show similar Raman spectroscopy in the 300 $cm^{-1}$–2000 $cm^{-1}$ region [9,10]. Hence, it is not easy to directly distinguish the same mineral salt TCM without combining analysis models in this fingerprint range. The low-wavenumber Raman spectrum (0–300 $cm^{-1}$) is very sensitive to the weak intermolecular interactions, skeleton vibration, and lattice vibration for materials, which are the intrinsic characteristics of solid-state structures [11]. Thus, definitive information about structural variations in minerals can be provided by low-wavenumber Raman spectral variations.

Terahertz (THz) waves usually refer to the electromagnetic waves between microwave and infrared bands, with a frequency ranging from 0.1 THz to 10 THz (3.3 $cm^{-1}$–330 $cm^{-1}$), and have the unique characteristics of being low-energy, label-free, and non-ionizing. THz waves can characterize low-frequency crystalline vibrations, hydrogen-bonding stretches, torsion vibrations, and molecular rotations at the molecular level [12,13]. THz waves are also sensitive to low-frequency collective vitiation, which is relevant to entire molecular and environment elements [14]. In particular, the THz spectroscopy technique has been applied to numerous fields, such as biomedicine detection [15–17], food safety monitoring [18–20], and pharmaceutical identification [21–23]. However, due to the complexity of the chemical constituents of TCMs, the absorption characteristic peak is not obvious, and sometimes it is even submerged. Overall, Raman spectroscopy is usually used to make a final confirmation of analytes of interest, and the information provided by THz spectroscopy can facilitate the analyte identification. Raman spectroscopy and THz spectroscopy have their strong points, but also shortcomings in investigating this spectral region. Thus, comprehensive chemical, structural, and environment information for analytes of interest can be simultaneously obtained from the combination of low-wavenumber Raman and THz spectroscopy.

In this work, an identification method of easily confused mineral TCMs is proposed by combining low-wavenumber Raman and THz spectroscopy. Six kinds of easily confused mineral TCMs, including coral skeleton, ophicalcitum, calamine, matrii sulfas exsiccatus, gypsum, and alumen, were measured using Raman microscope spectrometer and THz time-domain spectroscopy (THz-TDS). It was clearly observed that these mineral TCMs have different Raman characteristic spectroscopies in the 0–300 $cm^{-1}$ region. The THz spectra of these mineral TCMs showed that different kinds of mineral TCMs have different absorption coefficients in the 0.3–2.0 THz range. Furthermore, ophicalcitum and matrii sulfas exsiccatus were distinguished as having different processing methods based on THz absorption spectroscopy, whereas there were no obvious differences in Raman spectroscopy. The results indicate that the combination of low-wavenumber Raman and THz spectroscopy shows good feasibility for the identification of mineral TCMs.

## 2. Materials and Methods

### 2.1. Materials

Six kinds of mineral TCM, including coral skeleton, ophicalcitum, calamine, matrii sulfas exsiccatus, gypsum, and alumen, were purchased from a pharmacy (Tianjin, China) and used as received without further purification, as shown in Figure 1. These samples were all white or pale white crystalline powder and had no particular odor. It was hard to distinguish them by appearance. All samples are common drugs in clinics. Keleton, ophicalcitum, and calamine are all categorized as carbonates since their major components all include the $CO_3^{2-}$ group. Matrii sulfas exsiccatus, gypsum, and alumen are grouped as sulfates because they all have the $SO_4^{2-}$ anion group in the main chemical composition. The classification and major components of these TCMs are shown in Table 1. All samples were pulverized into powder by a pulverizer. The powder was filtrated through a 200-mesh sieve to minimize the scattering effects from sample particles during spectral measurements. The mixtures of the samples and high-density polyethylene (Sigma-Aldrich, Shanghai, China) were obtained with a mixing ratio of 1:1. Each kind of mixed powder was poured into a steel die and subjected to 6 MPa pressure for 3 min. Then it was made into circular tablets with a diameter of 10 mm and a thickness of 1.0 mm. Five replicates of each sample were prepared for reproducibility assessment.

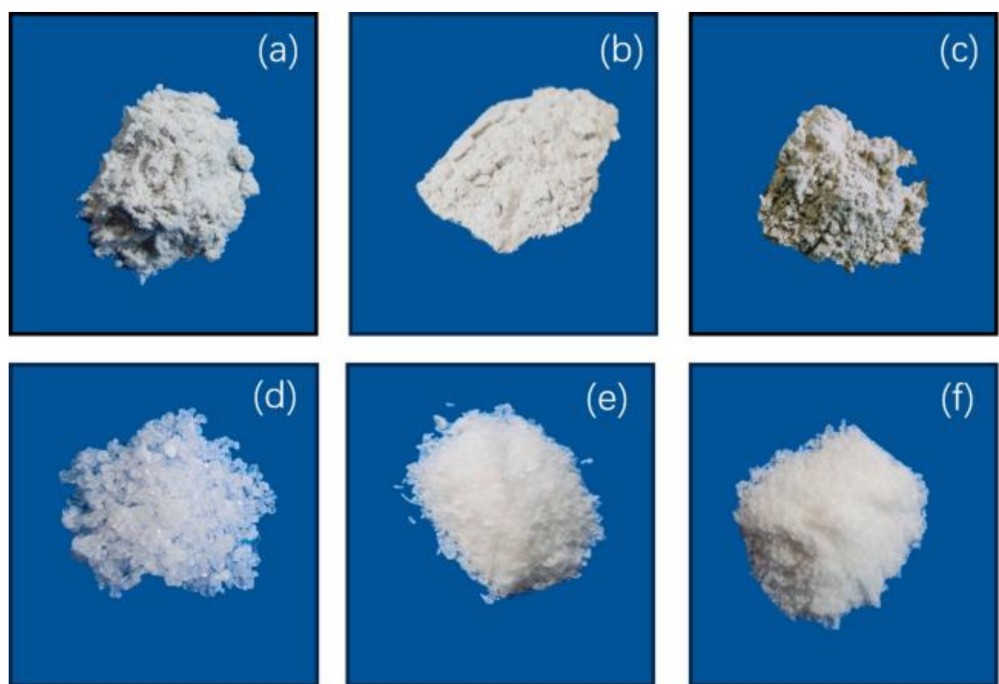

**Figure 1.** Pictures of (**a**) coral skeleton, (**b**) ophicalcitum, (**c**) calamine, (**d**) matrii sulfas exsiccatus, (**e**) gypsum, and (**f**) alumen.

**Table 1.** Classification and main components of mineral TCMs.

| Classification | Sample | Major Components |
|---|---|---|
| | Coral skeleton | $CaCO_3$ and $CaMg(CO_3)_2$ |
| Carbonate | Ophicalcitum | $CaCO_3$ |
| | Calamine | $ZnCO_3$ |
| | Matrii sulfas exsiccatus | $Na_2SO_4$ |
| Sulfate | Gypsum | $CaSO_4 \cdot 2H_2O$ |
| | Alumen | $KAl(SO_4)_2 \cdot 12H_2O$ |

*2.2. Apparatus and Procedure*

The Raman measurement was carried out with a commercial THz-Raman spectroscopy microscope system (Ondax, SureBlock$^{TM}$ Inc. Monrovia, CA, USA). The system was equipped with a single-mode stabilized diode laser module (Ondax SureBlock$^{TM}$ series, Inc. Monrovia, CA, USA), a spectrometer with 1200 g/mm grating (iHR-320, Horiba, Japan), and a microscope (Leica DM2500, Wetzlar, Germany). Backscattering light from the sample was collected by a 10× objective lens and filtered through a series of volume holographic grating (VHG) filters, and then focused into a spectrometer via a fiber-optic cable and detected by a CCD (Syncerity, Horiba, Japan, cooled to −50 °C). The initial spectra were acquired over a spectral range of −200–2000 cm$^{-1}$ with 2.5 cm$^{-1}$ resolution. Each Raman spectrum was recorded using 785 nm excitation with approximately 70 mW laser power and 10 s acquisition time. The final Raman spectra of each kind of mineral TCM were the averaged result of the five samples, where the spectrum of each sample came from three tests at different positions. The experimental temperature was maintained at room temperature during the whole experiment.

In order to acquire THz spectra of the samples, a commercially available THz time-domain spectrometer was used (Advantest Corp., TAS7500SP, Tokyo, Japan). The spectrum was measured from 0.1 to 2.0 THz, with a frequency resolution setting at 7.6 GHz and a dynamic range of 70 dB [24]. All the measurements were performed in transmission configuration at room temperature. The entire THz spectroscopy system was enclosed in a sealed box filled with dry air to reduce the water absorption effect of THz waves in the air. The relative humidity was always kept below 3% during the spectral measurements. THz spectrum measurement for each sample was repeated 5 times. Reference and sample signals were obtained in the absence and presence of the sample tablets, respectively. After being recorded by a detector, the THz time-domain waveforms of the reference and sample measurements were Fourier transformed to acquire the frequency-dependent parameters [25].

## 3. Results

The Raman spectra of coral skeleton, ophicalcitum, and calamine in the 0–1500 cm$^{-1}$ region are shown in Figure 2. As seen in Figure 2a, the coral skeleton exhibited four strong Raman peaks at 152, 205, 707, and 1086 cm$^{-1}$ and a relatively weak peak at 181 cm$^{-1}$. As seen in Figure 2b, the ophicalcitum had four strong characteristic peaks at 157, 283, 714, and 1088 cm$^{-1}$. Comparatively, calamine had four peaks at 156, 283, 712, and 1090 cm$^{-1}$, where the peak at 1090 cm$^{-1}$ showed a very strong Raman characteristic but the peak at 712 cm$^{-1}$ was relatively weak, as shown in Figure 2c. The Raman spectra of coral skeleton and ophicalcitum were published earlier, but are presented again for a reliable comparison with the results obtained from the other spectroscopic method [26]. The characteristic peak assignment of the three carbonates are listed in detail in Table 2 [26]. The Raman vibrational modes of the carbonate at or near 712 and 1086 cm$^{-1}$ arose from in-plane bending and symmetric stretching of the $CO_3^{2-}$ group, respectively. The characteristic peaks in the 0–300 cm$^{-1}$ range were assigned to lattice vibration within crystal. It is clear that the three carbonates had similar spectral properties in the 300–1500 cm$^{-1}$ region, which can be attributed to the same $CO_3^{2-}$ anionic group. The slight deviation in Raman vibrational mode above 300 cm$^{-1}$ was derived from the different cations. Therefore, it was hard to rapidly distinguish among the three carbonate mineral TCMs in the 300–1500 cm$^{-1}$ range. However, the shapes, peak positions, and intensities of the Raman spectrum for coral skeleton below 300 cm$^{-1}$ showed significant differences with ophicalcitum and calamine, which is due to the difference in the lattice structure. The coral skeleton belongs to an orthorhombic system whereas calamine and ophicalcitum are trigonal systems. Thus, the low-wavenumber Raman spectrum can be used to rapidly identify the carbonate mineral TCMs with different lattice structures. Besides the lattice vibration, the vibrational modes below 300 cm$^{-1}$ also reflect the inter-molecular and intra-molecular sample. The more accurate vibrational mode below 300 cm$^{-1}$ will be calculated by the theoretical calculation

in future work. For the ophicalcitum and calamine, the Raman peaks were still almost the same even below 300 cm$^{-1}$. Hence, we should find another method to distinguish the TCMs with similar lattice structures.

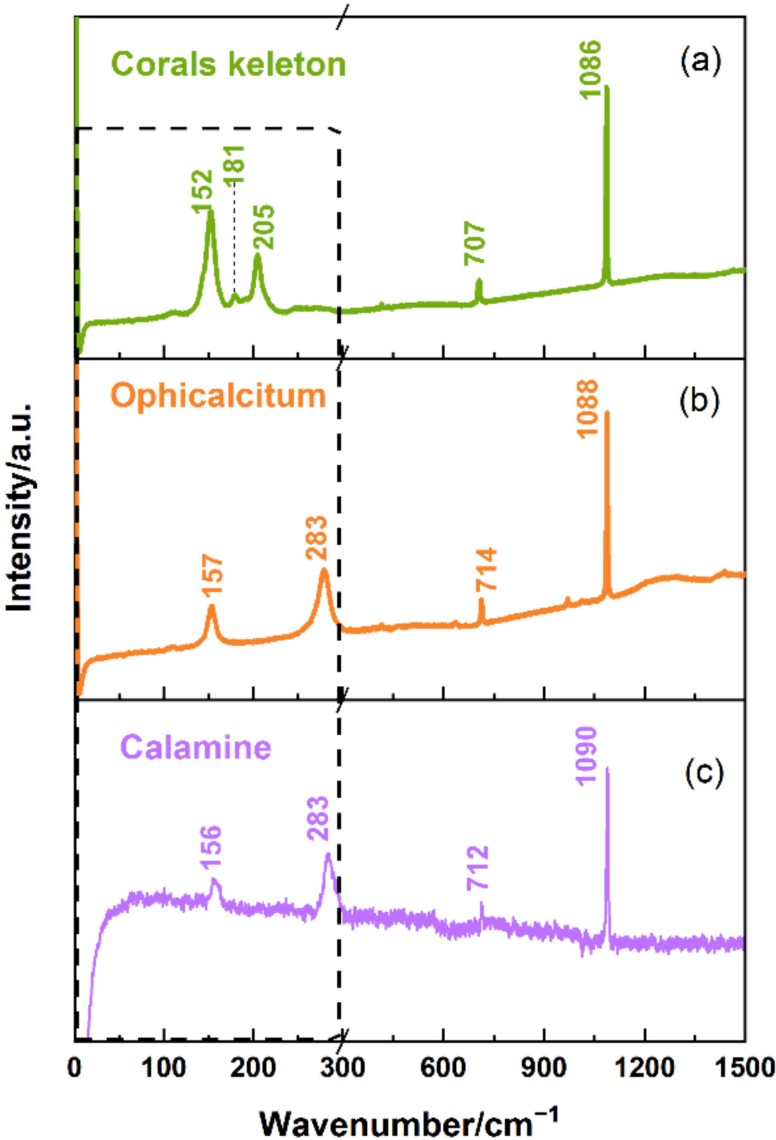

**Figure 2.** The Raman spectra of (**a**) coral skeleton, (**b**) ophicalcitum, and (**c**) calamine.

**Table 2.** The characteristic peak assignment of the three carbonate mineral TCMs.

| Peak Position/cm$^{-1}$ | | | Mode Assignments |
|---|---|---|---|
| **Coral Skeleton** | **Ophicalcitum** | **Calamine** | |
| 152,181,205 | 157,283 | 156,283 | Lattice vibration |
| 707 | 714 | 712 | $CO_3^{2-}$ in-plane bending |
| 1086 | 1088 | 1090 | $CO_3^{2-}$ symmetric stretching |

The THz spectra of carbonates including coral skeleton, ophicalcitum, and calamine were measured in 0.3–2.0 THz, as shown in Figure 3. It can clearly be seen that the absorption coefficients of these three kinds of TCMs were all increased with the increase in THz frequency. There was no obvious peak in the absorption spectra. The THz absorption coefficient of calamine was higher than that of coral skeleton in the 0.3–1.2 THz range,

where the absorption coefficient difference decreased with the increase in THz frequency. Especially, the absorption coefficients of calamine and coral skeleton almost overlapped in the range of 1.2–2.0 THz. In addition, the absorption coefficient of calamine was significantly higher than that of ophicalcitum in the entire 0.3–2.0 THz range, and the absorption coefficient of coral skeleton was higher than that of ophicalcitum in the 0.8–2.0 THz range. In other words, the three kinds of carbonate mineral TCMs had different THz spectra with the parameter of the absorption coefficient. Thus, the THz spectroscopy technique can be used as a complement to Raman spectroscopy for analyst identification.

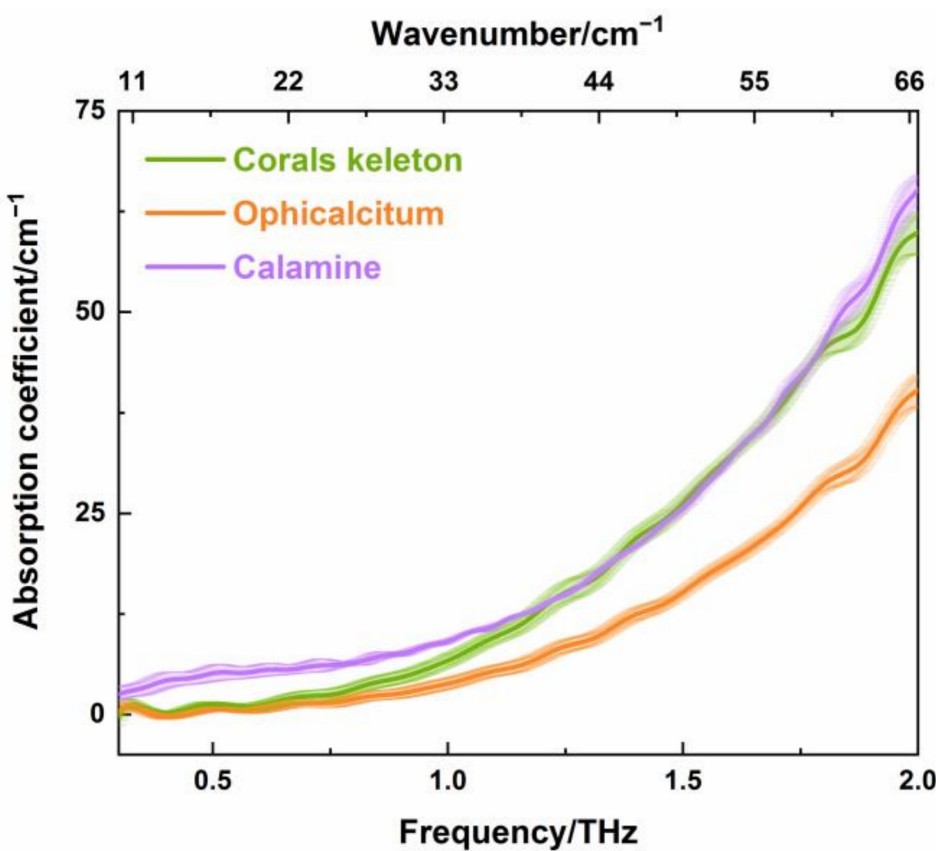

**Figure 3.** THz absorption spectra of coral skeleton, ophicalcitum, and calamine.

Then, three kinds of sulfate mineral TCM, including matrii sulfas exsiccatus, gypsum, and alumen, were measured, which have the same $SO_4^{2-}$ anionic. Figure 4 shows the Raman spectra of matrii sulfas exsiccatus, gypsum, and alumen. For matrii sulfas exsiccatus, there were three strong and sharp Raman characteristic peaks at 1104, 1137, and 1153 cm$^{-1}$ and one very strong peak at 990 cm$^{-1}$ at 300–1500 cm$^{-1}$. The Raman peaks at 452 and 467 cm$^{-1}$ and 621, 633, and 648 cm$^{-1}$ showed split characteristics. For gypsum, it exhibited three strong Raman characteristic peaks at 415, 495, and 1137 cm$^{-1}$; one very strong peak at 1010 cm$^{-1}$, and two relatively weak peaks at 619 and 672 cm$^{-1}$ in the range of 300–1500 cm$^{-1}$. For alumen, the Raman spectrum had three weak peaks at 456, 617, and 1135 cm$^{-1}$, a narrow strong characteristic peak at 933 cm$^{-1}$, and one split peak at 977 cm$^{-1}$. The broadband at the center frequency of 1240 cm$^{-1}$ arose from the slide underneath the sample. The entire vibrational mode assignment for Raman characteristic peaks of the three sulfates is summarized in Table 3 [26,27]. The Raman peaks at 300–1500 cm$^{-1}$ can be assigned to four vibration modes, i.e., symmetric stretching modes, symmetric bending modes, antisymmetric stretching modes, and antisymmetric bending modes [9,26]. It is possible that the cations distort the molecular structure and reduce the molecular symmetry or antisymmetry, which could cause the splitting of the associated characteristic Raman

peaks. Although the Raman spectra of matrii sulfas exsiccatus, gypsum, and alumen showed a little difference in the 300–1500 cm$^{-1}$ range, they are still easy to be confused. The differences in Raman spectra between the three sulfates at 300–1500 cm$^{-1}$ may come from the attribution of other minor components. Furthermore, a comparison of Raman spectra in the 0–300 cm$^{-1}$ range was analyzed. It was seen that matrii sulfas exsiccatus had four obvious characteristic peaks at 61, 82, 130, and 161 cm$^{-1}$; gypsum showed seven Raman bands at 91, 110, 123, 134, 147, 165, and 181 cm$^{-1}$; and alumen exhibited five relatively weak peaks at 38, 80, 120, 153, and 191 cm$^{-1}$. Most noticeably, different sulfate mineral TCMs had different low-wavenumber spectra, especially in the position of characteristic peaks. This is because different sulfate minerals have different lattice structures, which results in different lattice vibrations. Thus, the sulfate mineral TCMs can be effectively distinguished by low-wavenumber Raman characteristic peaks in the 0–300 cm$^{-1}$ region.

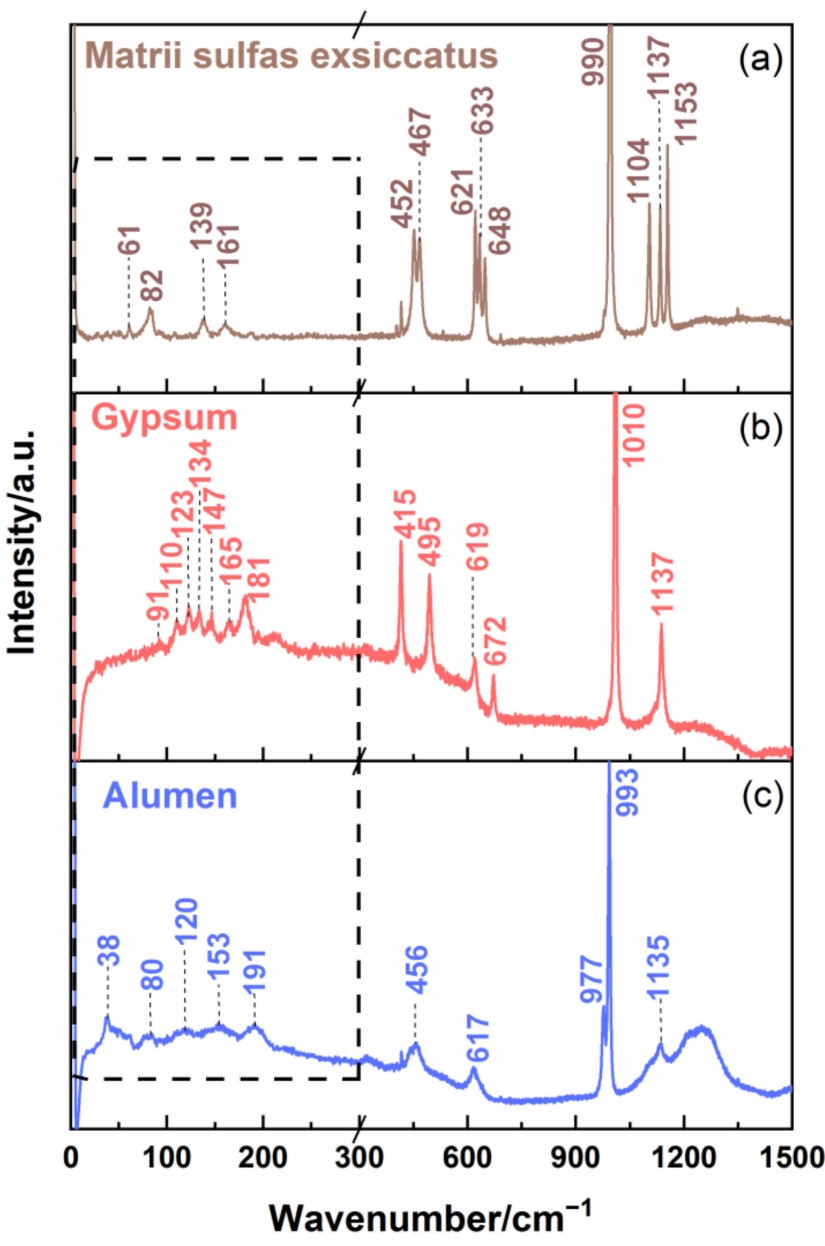

**Figure 4.** The Raman spectra of (**a**) matrii sulfas exsiccatus, (**b**) gypsum, and (**c**) alumen.

**Table 3.** Vibrational mode assignments for the Raman peaks of the sulfates.

| Peak Position/cm$^{-1}$ | | | Mode Assignments |
|---|---|---|---|
| **Matrii Sulfas Exsiccatus** | **Gypsum** | **Alumen** | |
| 61, 82, 139, 161 | 91, 110, 123, 134, 147, 165, 181 | 38, 80, 120, 153, 191 | Lattice vibration |
| 452, 467 | 415, 495 | 456 | SO$_4^{2-}$ symmetric bending |
| 621, 633, 648 | 619, 672 | 617 | SO$_4^{2-}$ deformation bending |
| 990 | 1010 | 977, 993 | SO$_4^{2-}$ symmetric stretching |
| 1104, 1137, 1153 | 1137 | 1135 | SO$_4^{2-}$ antisymmetric stretching |

Additionally, the THz spectra of matrii sulfas exsiccatus, gypsum, and alumen are shown in Figure 5. It can be seen that the absorption coefficients of these kinds of TCMs increased with the increase in THz frequency. For alumen, there were four absorption peaks at 1.05 THz (35.0 cm$^{-1}$), 1.66 THz (55.3 cm$^{-1}$), 1.79 THz (59.6 cm$^{-1}$), and 1.83 THz (60.9 cm$^{-1}$), which could be attributable to the low-energy lattice vibration, the molecular vibration caused by skeleton vibration, or rotation. More specifically, the THz absorption peak at 1.05 THz coincided with the Raman peak at 38 cm$^{-1}$. Here, it should be mentioned that a difference of several wavenumbers between THz and Raman mode frequency is allowed owing to the influence of factor group splitting [28,29]. Therefore, we can infer that the resonance absorption is induced by the lattice vibration, which was both infrared and Raman active at this characteristic peak. Moreover, the absorption peaks at 1.66 THz (55.3 cm$^{-1}$), 1.79 THz (59.6 cm$^{-1}$), and 1.83THz (60.9 cm$^{-1}$) might have been caused by collective molecular vibration and/or rotation, which cannot be indicated by the Raman spectrum of alumen. For matrii sulfas exsiccatus, it had one THz absorption peak at 1.87 THz (62.3 cm$^{-1}$), which corresponded to the Raman peak at 61 cm$^{-1}$. This means that the lattice vibration mode of matrii sulfas exsiccatus around 61 cm$^{-1}$ was not only infrared active, but also Raman active. Meanwhile, it is obvious that the absorption coefficients for the three kinds of sulfate were different in the range of 0.3–2.0 THz and followed the order of alumen > matrii sulfas exsiccatus > gypsum. Thus, sulfate mineral TCMs can be distinguished by THz absorption spectroscopy.

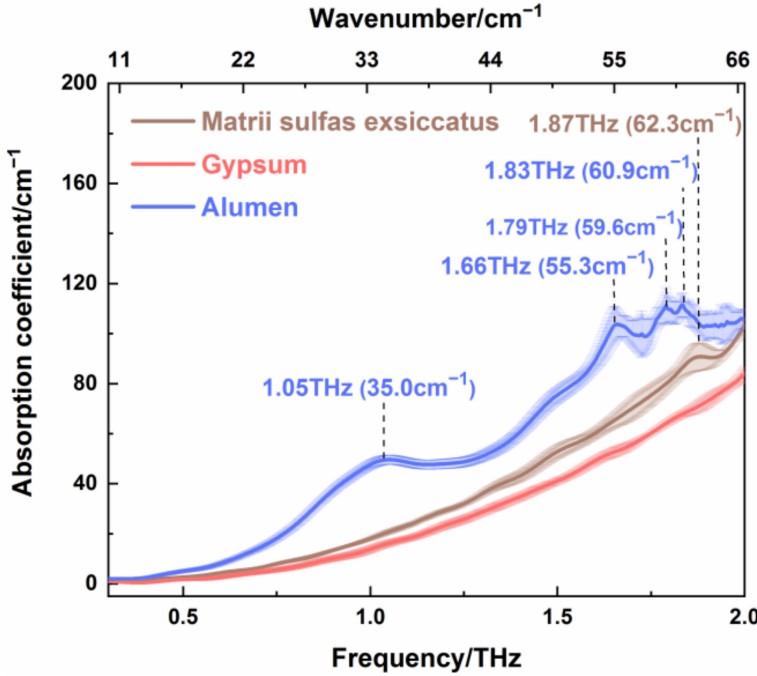

**Figure 5.** The THz absorption spectra of matrii sulfas exsiccatus, gypsum, and alumen.

Finally, two kinds of carbonate and sulfate mineral TCMs with different processing methods were measured using Raman and THz spectra. Calcination is a kind of processing method of TCM that can change the original traits of TCM and make it more suitable for clinics. Ophicalcitum and its calcined product, as shown in Figure 6a,b, were tested with Raman spectroscopy. Figure 6a is the same as Figure 1b for the comparison of the appearance of the ophicalcitum and the calcined product. Figure 6c,d show the Raman spectra of ophicalcitum and its calcined product in the 0–1500 cm$^{-1}$ region, respectively. It can clearly be seen that the characteristic peak, shape, and peak intensities of the spectra did not obviously change before and after calcination except for the rising baseline. This may be due to the impurities contained in calcined ophicalcitum. The small deviations in the two Raman peaks between ophicalcitum and its calcined product in the 300–1500 cm$^{-1}$ range were only 2 cm$^{-1}$, which is within the range of the allowable error of the system and can be ignored here. Thus, we can deduce that both the main chemical composition and lattice structure were not changed before or after calcination for ophicalcitum.

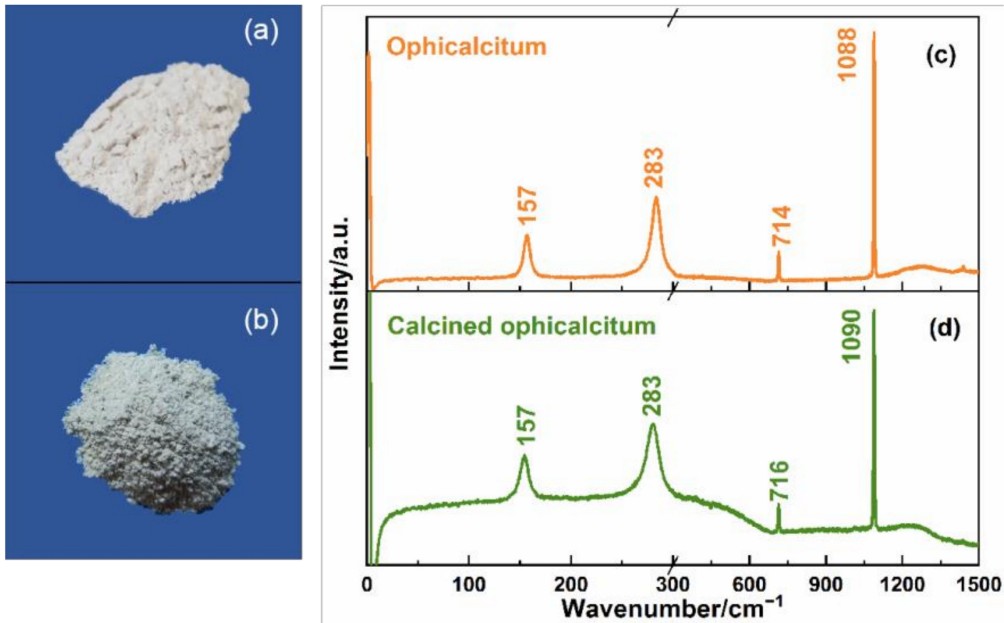

**Figure 6.** (**a**) Ophicalcitum, (**b**) calcined ophicalcitum, and the Raman spectra of (**c**) ophicalcitum and (**d**) calcined ophicalcitum.

Matrii sulfas exsiccatus is a popular kind of TCM and has long history in clinical treatment. Generally, matrii sulfas exsiccatus is prepared by the dehydration or weathering of mirabilite, the main constituent of which is $Na_2SO_4 \cdot 10H_2O$. Apart from being usable in medicine, mirabilite is an important raw material in industrial fields, such as papermaking, rubber, and textile. Although the main constituent is the same as $Na_2SO_4$, industrial-grade mirabilite cannot be used as a drug due to the different processing methods. Considering that matrii sulfas exsiccatus and industrial-grade mirabilite are both white crystalline in appearance, as shown in Figure 7a,b, they are easy to confuse in practical applications. Here, it should be mentioned that Figure 7a is the same as Figure 1d in order to demonstrate a clearer comparison of matrii sulfas exsiccatus and industrial-grade mirabilite. Figure 7c,d show Raman spectra of matrii sulfas exsiccatus and industrial grade mirabilite, respectively. It was found that there was also no distinct difference between them in characteristic peaks and shape except for the baseline, where the Raman spectrum baseline could have been caused by small amounts of impurities in the industrial-grade mirabilite sample.

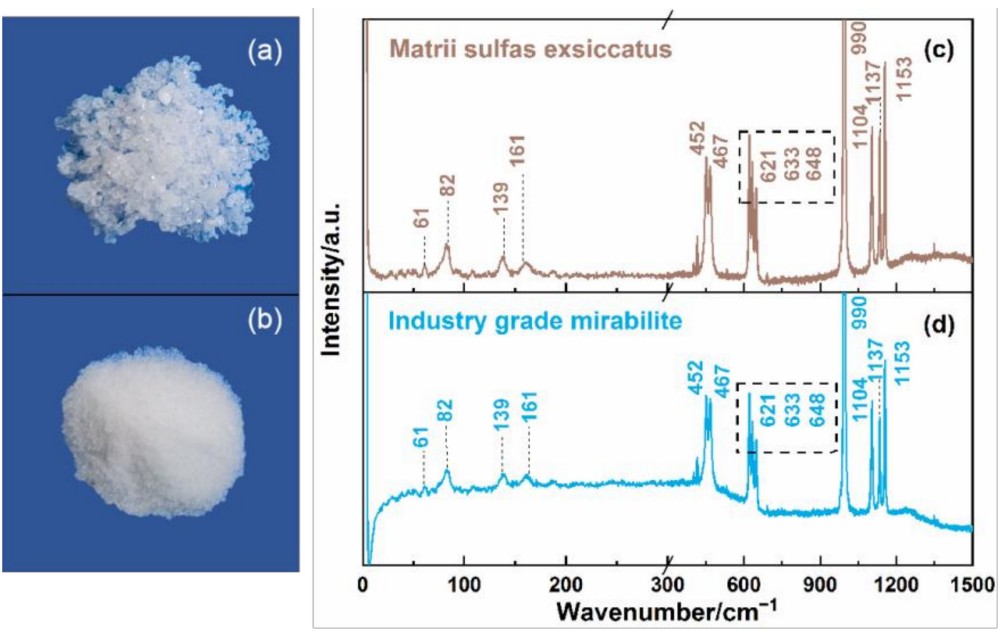

**Figure 7.** (**a**) Matrii sulfas exsiccatus, (**b**) industrial grade mirabilite, and the Raman spectra of (**c**) matrii sulfas exsiccatus and (**d**) industrial grade mirabilite.

Figure 8 shows the THz absorption spectra of the samples measured in Figures 6 and 7. It can be seen in Figure 8a that the THz spectrum of ophicalcitum had no significant peak. The THz absorption coefficient of ophicalcitum was consistent with the calcined ophicalcitum in the range of 0.3–1.0 THz, whereas there was a marked difference in the 1.0–2.0 THz region. More specifically, the difference in absorption coefficient gradually increased with the increase in THz frequency. This was probably due to particle scattering at high THz frequencies. Thus, we can deduce that THz absorption spectrum can distinguish ophicalcitum from its calcined product directly at a higher THz frequency. Figure 8b exhibits the THz spectra of matrii sulfas exsiccatus and industrial-grade mirabilite. Compared to the industrial-grade mirabilite, the matrii sulfas exsiccatus showed the same absorption coefficient in the 0.3–0.7 THz range, whereas it had a higher absorption coefficient at 0.7–2.0 THz. It was also observed that the THz spectrum of matrii sulfas exsiccatus had one characteristic peak at 1.87 THz (62.3 $cm^{-1}$), whereas the industrial-grade mirabilite showed an absorption peak at 1.92 THz (63.9 $cm^{-1}$). Combined with the Raman spectra in Figure 7c,d, the two THz absorption peaks of matrii sulfas exsiccatus and industrial-grade mirabilite both corresponded to the Raman peak at 61 $cm^{-1}$. This means that different processing methods cannot change the infrared and Raman active mode of mirabilite. However, the environmental information of the analyst, such as water content, purity, or grain size, would be changed after different processing methods. This would cause the THz absorption peak and the absorption coefficient to be changed. Therefore, the THz spectrum can provide valuable additional information about TCMs with different processing methods that cannot be inferred from Raman analysis. It can make up for the deficiency of Raman spectroscopy.

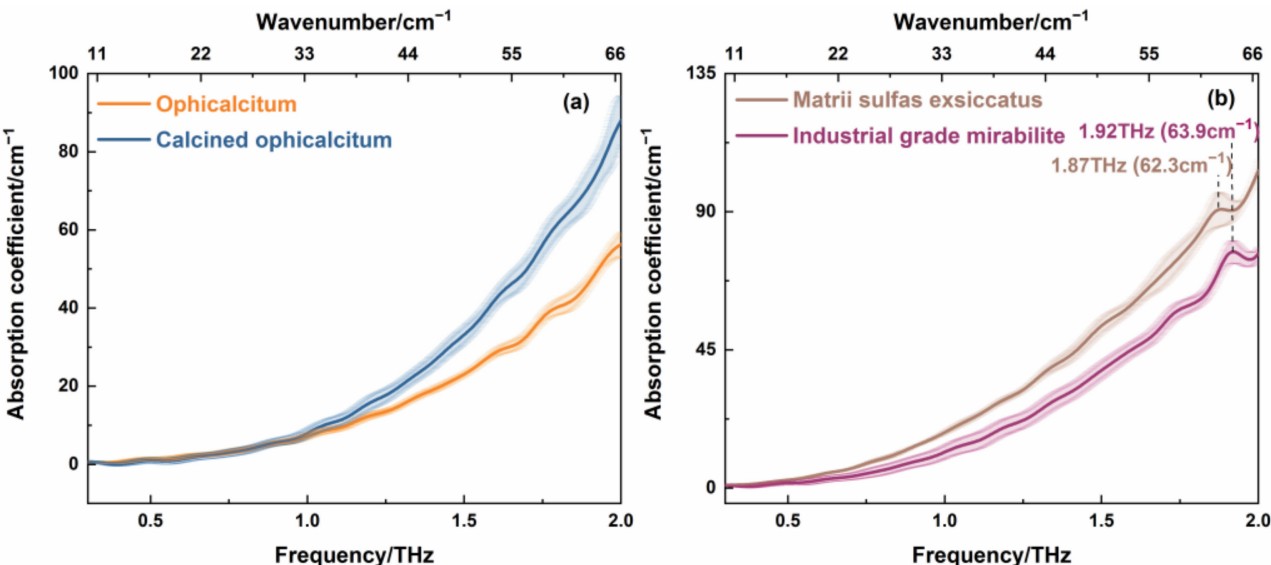

**Figure 8.** The THz absorption coefficients of (**a**) ophicalcitum and its calcined product, and (**b**) matrii sulfas exsiccatus and industrial-grade mirabilite.

## 4. Conclusions

In conclusion, the combination of low-wavenumber Raman and THz spectroscopy to distinguish of easily confused mineral TCMs has been demonstrated in this paper. The experiment results show that six kinds of confused mineral TCMs have significantly different Raman characteristic peaks in the 0–300 cm$^{-1}$ range. The mineral TCMs of carbonates or sulfates can also be distinguished based on the THz absorption coefficient in the 0.3–2.0 THz range. The THz absorption spectroscopy can especially discriminate ophicalcitum and matrii sulfas exsiccatus with different processing methods, making up for the limitation of low-wavenumber Raman spectroscopy. Thus, the combination of THz and low-wavenumber Raman spectroscopy techniques can provide much richer information on mineral TCMs. This approach shows an important role in evaluating the quality and safety of mineral TCM.

**Author Contributions:** Conceptualization, M.G. and Y.W. (Yuye Wang); methodology, M.G.; software, H.L.; validation, Y.W. (Yu Wang), C.H. and J.Z.; formal analysis, M.G.; investigation, M.G.; resources, D.X.; data curation, H.L.; writing—original draft preparation, M.G.; writing—review and editing, Y.W. (Yuye Wang); visualization, M.G.; supervision, B.W.; project administration, J.Y.; funding acquisition, Y.W. (Yuye Wang). All authors have read and agreed to the published version of the manuscript.

**Funding:** This work was funded by the National Natural Science Foundation of China (NSFC) under grant No. U1837202, No. 62175182, and No. 62011540006.

**Institutional Review Board Statement:** Not applicable.

**Informed Consent Statement:** Not applicable.

**Data Availability Statement:** The data that support the findings of this study are available from the first author and the corresponding author upon reasonable request.

**Conflicts of Interest:** The authors declare no conflict of interest.

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
