# Peer review of "Rapid Identification of Easily-Confused Mineral Traditional Chinese Medicine (TCM) Based on Low-Wavenumber Raman and Terahertz Spectroscopy"

_photonics, doi:10.3390/photonics9050313_

Round 1

Reviewer 1 Report

The work contributed by the authors is important and interesting. I would like to recommend the work to be published in Photonics by addressing the following questions. 1. By comparing the images of mineral TCMs in Fig. 1 and those in Fig.6 and Fig.7., I suggest the authors to check the pictures, and make sure the pictures were used in the right way. 2. The pictures in Fig.6a, Fig.6b and Fig.7a are the same as Fig.1e, 1d and 1b, correspondingly. This is inappropriate. 3. In 2.2 Apparatus and Procedure, the authors wrote “The Raman measurement was carried out with a commercial THz-Raman spectroscopy microscope system.” Please add necessary information for the brand and manufacturer.

Reviewer 2 Report

This article focuses on the application of two different low-frequency vibrational spectroscopic techniques for the identification of various traditional Chinese medicines. While the topic is timely and interesting, the scope of the article needs to be expanded beyond simple characterization as in many instances it is not directly obvious that the use of THz region would actually benefit such analysis. Additionally, there are other things that need to be addressed before publishing:

  1. Firstly, from simple characterization of the samples, it is very hard to deduct the usefulness of the application of such techniques without showcasing some real examples either from qualitative or quantitative analysis perspective. Authors themselves state that in some instances THz region displays similar features among samples. Additionally, as typical per most inorganic substances, the typically advantageous high signal propensity of the phonon modes is also not present. Thus, considering the availability of the data from both Raman regions, I would recommend authors to conduct additional studies, for example, by examining artificial mixtures of the compounds and comparing the performance of the models (for example, from PCA or PLS etc.) using different spectral data. Also, authors must be very careful for proposing the use of THz absorption data that only differ in its baseline. As this technique is very sensitive to water, it is very speculative that it could be used efficiently without the presence of distinct spectral features (this aspect could also be tested by sorbing different amounts of water to the samples and evaluating the spectral changes).
  2. Clarification about the lattice mode assignments needs to be presented. Without the use of theoretical calculations (such as periodic DFT) or existing information in the literature, one can’t simply denote all the modes below certain wavenumber threshold as lattice vibrations. While, generally, spectral region below 300 cm-1 is considered to encompass them, the continuum between intermolecular and intramolecular vibrational modes can still differ among different analytes.
  3. Some technical editing changes are also necessary:
  • Chemical formula denotations need to be correctly presented within the text (e.g., SO42- vs SO42-);
  • Inconsistent use of the first uppercase letter and punctuation for figure labels;
  • Page 2 line 49: exists needs to be replaced with displays (or similar);
  • Page 2 lines 69-70: sentence has a weird structure, please revise;
  • Page 2 line 76: sentence (or paragraph?) is incomplete!
  • Page 2 line 93: 10mm -> 10 mm;
  • Page 3 line 108: I would recommend rewording “volume holographic grating filters” to “holographic volume Bragg gratings”;
  • Page 3 line 111: superscript is missing for wavenumbers (cm-1);
  • Page 4 line 129: 0-1500cm-1 -> 0-1500 cm-1;
  • Page 4 line 143: same as above;
  • Page 6 line 188: same as above;
  • Page 8 line 242: same as above.

Round 2

Reviewer 2 Report

Overall, authors have addressed most of the raised issues. However, there are still number of technical formatting things that need to be fixed (i.e., spaces between the numbers and the units etc.). Additionally, the use of English language needs to be checked as many of the added (as well as existing) paragraphs contain numerous grammar errors that are laborious to list here, and perhaps some type of proofing services need to be used for the finalization of the manuscript.  

Author Response

Response 1: We gratefully appreciate your comments. We have carefully checked the full manuscript, and fixed technical formatting things, including spaces between the numbers and the units. In addition, we corrected the grammatical errors and made an effort to correct the spelling and grammar errors and polish the whole manuscript. We would to confirm that the suitably revised manuscript is understandable to readers. And the amendments have been highlighted using red colored in the revised manuscript. For example:

  1. Keleton, ophicalcitum, and calamine are all categorized as carbonate since their major components all include the CO32-

  1. Then it was made into circular tablets with the diameter of 10 mm and the thickness of 1.0 mm.

  1. But it still displays some shortcomings, such as being highly sensitive to the effect of water, and difficulty with data modeling.

  1. Although the main constituent is the same as Na2SO4, industrial grade mirabilite cannot be used as drug due to different processing methods.

  1. Considering matrii sulfas exsicctaus and industrial grade mirabilite are both white crystalline in appearance, as shown in Fig. 7(a) and (b), they are easy to confuse in practical application.
